# Mural Cells: Potential Therapeutic Targets to Bridge Cardiovascular Disease and Neurodegeneration

**DOI:** 10.3390/cells10030593

**Published:** 2021-03-08

**Authors:** Alexander Lin, Niridu Jude Peiris, Harkirat Dhaliwal, Maria Hakim, Weizhen Li, Subramaniam Ganesh, Yogambha Ramaswamy, Sanjay Patel, Ashish Misra

**Affiliations:** 1Heart Research Institute, Sydney, NSW 2042, Australia; Alexander.Lin@hri.org.au (A.L.); npei3711@uni.sydney.edu.au (N.J.P.); Harkirat.Dhaliwal@hri.org.au (H.D.); Maria.Hakim@student.uts.edu.au (M.H.); Eva.Li@hri.org.au (W.L.); sanjay.patel@hri.org.au (S.P.); 2School of Biomedical Engineering, Faculty of Engineering, The University of Sydney, Sydney, NSW 2006, Australia; yogambha.ramaswamy@sydney.edu.au; 3Sydney Medical School, The University of Sydney, Sydney, NSW 2006, Australia; 4School of Life Sciences, Faculty of Science, University of Technology Sydney, Sydney, NSW 2007, Australia; 5Department of Biological Sciences and Bioengineering, Indian Institute of Technology Kanpur, Kanpur, Uttar Pradesh 208016, India; sganesh@iitk.ac.in; 6The Mehta Family Centre for Engineering in Medicine, Indian Institute of Technology Kanpur, Kanpur, Uttar Pradesh 208016, India; 7Cardiac Catheterization Laboratory, Royal Prince Alfred Hospital, Sydney, NSW 2050, Australia; 8Faculty of Medicine and Health, The University of Sydney, Sydney, NSW 2006, Australia

**Keywords:** mural cells, pericytes, vascular smooth muscle cells, atherosclerosis, Alzheimer’s disease, cardiovascular disease, neurodegeneration

## Abstract

Mural cells collectively refer to the smooth muscle cells and pericytes of the vasculature. This heterogenous population of cells play a crucial role in the regulation of blood pressure, distribution, and the structural integrity of the vascular wall. As such, dysfunction of mural cells can lead to the pathogenesis and progression of a number of diseases pertaining to the vascular system. Cardiovascular diseases, particularly atherosclerosis, are perhaps the most well-described mural cell-centric case. For instance, atherosclerotic plaques are most often described as being composed of a proliferative smooth muscle cap accompanied by a necrotic core. More recently, the role of dysfunctional mural cells in neurodegenerative diseases, such as Alzheimer’s and Parkinson’s disease, is being recognized. In this review, we begin with an exploration of the mechanisms underlying atherosclerosis and neurodegenerative diseases, such as mural cell plasticity. Next, we highlight a selection of signaling pathways (PDGF, Notch and inflammatory signaling) that are conserved across both diseases. We propose that conserved mural cell signaling mechanisms can be exploited for the identification or development of dual-pronged therapeutics that impart both cardio- and neuroprotective qualities.

## 1. Introduction

The vascular system is a conduit which delivers vital nutrients, oxygen, and hormones throughout our body. As such, the development of the vascular system is a very precise process centered on the needs of the surrounding tissues. Fibroblast growth factor and vascular endothelial growth factor signaling are the predominant molecular signals for angiogenesis in utero [1,2]. Blood vessels primarily have three layers: the tunica intima, tunica media and tunica adventitia. Endothelial cells (ECs) line the luminal surface of the vessel wall, composing the innermost tunica intima layer. Smooth muscle cells (SMCs) are circumferentially arranged to make up the tunica media layer of all vessels except capillaries. Capillaries are not lined with SMCs but have a discontinuous layer of cells known as pericytes (Figure 1). The tunica adventitia is the outermost, supportive layer primarily composed of fibroblasts, extracellular matrix and progenitor cells [3].

The vascular system transports blood in a very specific manner. Blood leaves the heart via arteries to either arterioles of the pulmonary circulation or systemic circulation. Arterioles are the major source of resistance and therefore control of blood flow in the vascular system [4]. Arterioles supply blood to capillaries, the major site of exchange of gases, nutrients, and wastes. Capillaries feed into venules, then eventually veins before returning blood to the heart. Vascular SMCs (VSMCs) and pericytes, collectively referred to as mural cells, play an important role in the regulation of blood pressure, control of blood distribution and provision of structural support and integrity to the vessel wall [5]. As such, mural cell dysfunction plays a crucial role in the development of many issues pertaining to the vascular system.

VSMC dysfunction plays a crucial role in the pathogenesis and progression of a number of cardiovascular conditions. The classical SMC-centric case are atherosclerotic plaques, whose complications such as plaque rupture or embolism account for the leading cause of death globally. This process begins with a sub-intimal accumulation of modified lipoproteins and damage associated molecular patterns (DAMPs), leading to a low-grade inflammatory response. The pro-inflammatory environment leads to activation of several vascular cell types including VSMCs, ECs and macrophages and, subsequently, leads to the formation of an atherosclerotic plaque composed of a VSMC cap and a necrotic core [6]. Atherosclerotic plaque development and vulnerability is often described in terms of VSMC proliferation and transdifferentiation from a normal contractile phenotype to a synthetic one; the exact mechanisms underlying this process will be highlighted in subsequent sections [7]. Furthermore, a growing body of research has suggested that mural cell plasticity is far more important in disease pathogenesis than previously believed, as VSMC transdifferentiation can lead to atheroprotective or atheroprone phenotypes. In addition to cardiovascular conditions, mural cells also play a critical role in neurological conditions. There is growing evidence to show that vascular mural cell dysfunction, particularly pericyte dysfunction can lead to the onset of neurodegenerative diseases, such as Alzheimer’s Disease (AD) and Parkinson’s Disease (PD).

One of the earlier hypotheses in relation to this was described in 1993 in the “vascular hypothesis of AD”, whereby cerebrovascular abnormalities led to cerebral hypoperfusion and hypoxia subsequently leading to neurodegenerative changes [8]. Since then, numerous studies have bolstered this hypothesis and similar concepts have been explored in other neurodegenerative disease such as PD. In AD, a reduction of pericyte coverage in the cerebral microvasculature has been shown to impair BBB integrity and reduce Aβ clearance, thereby leading to a promotion of the AD disease process [9]. It has been postulated that pericytes play a crucial role in the spreading of the presynaptic protein α-synuclein (αSyn) deposits that is linked genetically and neuropathologically to PD [10]. In addition, numerous longitudinal and cross-sectional studies have confirmed this important “head-to-heart” association. For instance, the Rotterdam study, a large prospective population-based analysis, showed significantly poorer scores of cognitive functions (clinical markers of dementia) in patients with a history of prior cardiovascular incidents compared to those that had none [11].

In this review, we take a closer look at the global roles of mural cells in the pathogenesis of these cardiovascular and neurodegenerative diseases and some of the shared signaling mechanisms that could bridge the gap between cardiovascular disease and neurodegenerative disease. Identification of such conserved mural cell signaling pathways can aid in the development of dual-pronged therapeutics intended for patients with cardiovascular disease who may be at increased risk of neurological conditions such as dementia.

## 2. Mural Cells and Cardiovascular Disease

Cardiovascular disease (CVD) is an umbrella term used when referring to diseases affecting the heart and its subsequent blood vessels. They are particularly problematic as they are the highest cause of death worldwide. The World Health Organization estimates that CVDs were responsible for the death of nearly 18 million people (31% of global total) in 2016 [12]. CVDs are currently far more common in lower socioeconomic environments, as well as older individuals. Research in this field is needed as the incidence of CVDs is expected to rise in the near future due to an increase in life expectancy combined with more prevalent obesity, which is a well-known risk factor. Despite the global burden of CVDs, it has been suggested that good education, access to healthcare, and management of known risk factors could drastically reduce their mortality rate. One major cause of CVDs is atherosclerosis, which can lead to ischemia and stroke.

### 2.1. Atherosclerosis: A General Overview

Atherosclerosis is the build-up of lipids, cells and other cellular debris in the arteries, which form the characteristic atherosclerotic plaque [13,14]. This plaque has the potential to restrict blood flow directly, or indirectly via thrombus formation due to the rupture or erosion of the plaque, containing the necrotic thrombogenic core [15]. Due to the variance in location, size and composition of the plaque, atherosclerosis can cause different abnormalities throughout the body and as such is the most prominent cause of CVD [16,17]. For example, cerebral atherosclerosis can lead to strokes and cause neurodegeneration [18]. Atherosclerosis is believed to be caused by a combination of both environmental and genetic factors. Environmental factors include: smoking, high fat diets and sedentary lifestyles. Genome wide association studies have correlated numerous single nucleotide polymorphisms with disease susceptibility (such as *Cdkn2a, Cdkn2b, Lpa* and *Ldlr*) and highlighted the need to understand more about this debilitating disease [19]. One of the most prominent genes associated with atherosclerosis is apolipoprotein E (*Apoe*) and its corresponding protein is considered to be vital for the removal of lipoproteins that contain high amounts of cholesterol [20]. It has been extensively shown that *Apoe* mutations can lead to the development of plaques through various mice models combined with high fat diets [21]. Similarly, the low-density lipoprotein receptor (LDLR) also plays an important role in removal of lipoproteins in blood. As such, *Ldlr^-/-^* mice are also commonly used in atherosclerosis models, and may better mimic human pathogenesis than *Apoe^-/-^* mice [22].

On a molecular level, EC dysfunction causes the sub-intimal accumulation of lipoproteins, which are thought to act as DAMPs to activate an inflammatory response [6,23]. These affected ECs express adhesion molecules (such as: VCAM-1, ICAM-1 and P-Selectin) that allow the recruitment of circulating monocytes which thus differentiate into macrophages [23,24]. The accumulated low-density lipoproteins (LDLs) are then oxidized to form oxidized LDLs (oxLDLs) which are subsequently phagocytosed by the macrophages, which become foam cells [3]. Foam cells have the capacity to recruit more monocytes in a chain reaction that worsens disease progression [25]. This pro-inflammatory environment stimulates SMC migration from the medial wall, to form a protective fibrous cap over the plaque [6]. This process is summarized briefly in Figure 2.

### 2.2. VSMC Plasticity Plays an Important Role in the Progression of Atherosclerosis

VSMCs play a major role in the initial formation of the atherosclerotic plaque. Lineage tracing studies in murine embryos have shown that SMCs from differing vessels have different developmental origins [26]. It should therefore be no surprise that VSMCs from different developmental origins (e.g., the secondary heart field and neural crest) respond differently to cytokines and growth factors such as TGFβ1, PDGF-BB and these cells have different functional properties. This may explain the observation that some vessels are more prone to atherosclerosis than others [27,28]. Interestingly, clonal analysis using multicolor Cre reporter mice such as rainbow and confetti, which involves tracing of progenitor cell clones (identical progeny), has been used to identify that only one or two VSMCs undergo clonal expansion to form the fibrous cap and any smooth muscle derived cells within the plaque [7]. Therefore, what makes these VSMCs special? One potential answer is that there exists a small pool of stem cell antigen-1 (Sca1–a hematopoietic stem cell marker) marked VSMCs, that are more plastic and proliferative than other medial VSMCs, and therefore are responsible for the injury or inflammatory response [28,29,30,31,32]. It has been shown that these Sca1^+^ VSMCs are more proliferative and thus play a larger role in vessel repair than their regular VSMC counterparts [29]. The origin of these cells has not yet been fully established. They potentially arise from vascular stem cells in the adventitia, and/or induced under pathological conditions [28,29,32]. Given that not all the VSMC derived plaque cells are Sca1 positive and yet have a common VSMC progenitor, Dobnikar et al. postulate that Sca1 expression is modulated with VSMC phenotypic changes [28]. However, research into these Sca1^+^ SMCs is still new, and a lot more research is needed to understand their full potential and mechanisms in the disease state.

Plasticity refers to the ability of cells to modulate their phenotype under certain conditions. VSMCs and other plaque cells (ECs, macrophages) in atherosclerosis exhibit remarkable plasticity under hyperlipidemic conditions, and this is one of the reasons why atherosclerosis remains such a complicated disease. The plasticity of these cells also leads to a great deal of cellular heterogeneity, which has been identified by single cell RNA sequencing (sc-RNA seq) data [33,34]. The importance of non-VSMC plasticity (such as macrophage to mesenchymal transition (MMT) and endothelial to mesenchymal transition (EndoMT)) is the subject of recent growing research. For example, Newman et al. discuss the contribution and potential atheroprotective roles of EndoMT and MMT to fibrous cap ACTA2^+^ cells [35,36]. However, for the purposes of this review, we will solely discuss the plasticity of VSMCs. As mentioned previously, during the progression of atherosclerosis, we observe that the contractile medial VSMCs migrate and de-differentiate to a more synthetic phenotype to maintain the fibrous cap. Medial VSMCs in *Apoe^-/-^* mice under normal chow diet express smooth muscle myosin heavy chain (SMMHC) and α-SMA. In comparison, VSMCs in the fibrous cap of atherosclerotic vessels have downregulated SMMHC expression, and have collagen and matrix metallopeptidase expression–indicators of a more synthetic phenotype which is involved in plaque stability [37].

The heterogeneity of VSMCs is evident by research that has shown their ability to transdifferentiate into different cell types (according to markers), such as macrophage-like and osteochondrogenic-like states [7,38,39,40,41]. Some of these transdifferentiated states are atheroprotective whereas others are atheroprone and contribute to disease progression. Rong et al. made the seminal discovery that cholesterol loading could induce VSMC transdifferentiation to a macrophage-like state, shown by decreases in α-actin and α-tropomyosin, and subsequent increases in CD68 and Mac-2 expression. This idea was reinforced by the observation that these transdifferentiated VSMCs not only expressed macrophage markers, but also gained more phagocytic capabilities compared with regular VSMCs [38]. When discussing transdifferentiation, we must place emphasis on macrophage-like states, as expression of similar markers does not necessarily mean they have an equivalent function. Indeed, Vengrenyuk et al. showed that while these macrophage-like cells do indeed gain some phagocytic function, they still have a diminished capability compared with regular myeloid-derived macrophages [42]. Studies have suggested that around 50% of foam cells within the plaque are of VSMC origin, and that the downregulation of ATP-binding cassette transporter A1 (ABCA1) could be responsible for the cholesterol accumulation observed in vivo [40]. Despite this, while some macrophage markers are upregulated, there are still many genes (Mertk, Fcgr1) that remain in a VSMC-like state [42]. Nevertheless, VSMC to macrophage transdifferentiation is a serious consideration for any future therapies.

VSMCs also have the capability of transdifferentiating into osteochondrogenic-like cells which are thought to contribute to calcium phosphate deposition in the plaque. This crystallization contributes to the necrotic core, which increases the severity of atherosclerosis and likelihood of ischemia developing [39]. This process has been characterized by a loss in SM22α and α-actin, and an upregulation of osteopontin, cbfa1, osteocalcin and alkaline phosphatase [41]. Naik et al. were the first to use lineage tracing to identify that over 75% of osteochondrogenic-like cells in atherosclerotic regions were of SMC origin, with the rest being derived from the bone marrow [43].

However, the molecular mechanisms behind VSMC transdifferentiation to macrophage-like and osteochondrogenic-like states are not yet completely known. A potential mechanism underlying SMC plasticity is via the transcription factors Klf4 and Oct4, which are well-known for their role in inducing pluripotency [33]. While these factors are used in conjunction when producing induced pluripotent stem cells, it is interesting that this is not the case in atherosclerosis. Knocking out Klf4 in VSMCs appears to be beneficial, as it produced a more stable plaque, a smaller plaque size and a decrease in macrophage-like cells. ChIP-seq results indicate that Klf4 targets phagocytotic and inflammatory genes that are involved in atherosclerosis pathogenesis [44]. In contrast, Oct4 knockout in VSMCs appears to have the opposite effect, as seen by an increase in plaque size, and reduced VSMC migration. Interestingly, it was found that Oct4 expression may be dependent on Klf4 expression levels [45]. The importance of VSMC transdifferentiation was further verified by Alencar et al., who used sc-RNA seq to investigate the plaque heterogeneity, and found that of the 14 distinct cell clusters, 7 of them were derived from VSMCs. Crucially, the majority of VSMC derived cells did not express classical contractile markers. Knocking out Klf4 seemed to increase the VSMC contractile marker expressing clusters while decreasing the osteochondrogenic markers [33]. These results suggest that Klf4 and Oct4 may affect disease progression by modulating the transdifferentiation of VSMCs to different phenotypic cell types, and thus should be considered when discussing future therapeutic options.

### 2.3. PDGF Signaling in Atherosclerosis

While there are multiple signaling mechanisms responsible for the changes in VSMCs in atherosclerosis, only a couple will be briefly discussed in this review. Platelet derived growth factor (PDGF) signaling is utilized throughout the body for a number of biological functions including tissue repair and development [46]. It has been well established that PDGF is crucial for the proliferation and migration of VSMCs [47]. However, the role of PDGF signaling in cardiovascular diseases such as atherosclerosis is still not completely explored. The current belief is that PDGF plays a role in VSMC transition from the contractile to synthetic phenotype, which occurs during plaque formation. Sano et al. showed that PDGF receptor (PDGFR) β suppression caused a reduction in VSMC number and fibrous cap size [48]. Similarly, Wan et al. found that inhibiting PDGFR-β conserved the VSMC contractile phenotype which was associated with improved outcomes following subarachnoid hemorrhage [49]. More specifically to atherosclerosis, it has been shown that increasing PDGFR-β signaling causes VSMCs to release cytokines and promotes inflammation, overall accelerating disease progression [50]. As such, PDGF signaling presents a potential therapeutic target for atherosclerosis.

### 2.4. Notch Signaling in Atherosclerosis

In recent years the Notch signaling pathway has gained significant attention in the context of vascular biology and is currently being investigated in atherosclerosis. Notch signaling and its underlying mechanisms have already been extensively reviewed [51,52]. Very briefly, binding of a ligand to the Notch receptor results in the cleavage of the Notch intracellular domain (NICD), which is translocated to the nucleus where it alters gene expression. In mammals, there are four Notch receptors, and five main ligands–Delta-like ligands (Dll1, Dll3 and Dll4), and Jagged ligands (Jagged1 and Jagged2) [51]. Notch is highly conserved throughout numerous species and is known to play a critical role in development, as well as during tissue homeostasis. In the vasculature, ECs express Dll4, Jagged1, Notch1 and Notch4. VSMCs express Dll4, Jagged1, Notch1, Notch3 and Notch4. However, this is a large simplification, as ligand/receptor expression varies depending on location and vessel type [53].

The role of Notch in the vasculature during development has been demonstrated by a number of studies. By deleting the Notch binding domain of Jagged1, Xue et al. observed that homozygous mutants were embryonic lethal. These embryos exhibited hemorrhaging and vascular defects [54]. Similarly, *Notch1^-/-^* mice and *Notch1 Notch4* double knockout mice showed severe vascular defects, and Krebs et al. suggest Dll4 is responsible for activating both of these receptors during embryonic vasculature development [55]. Not only is Notch signaling required for development, but it is also required in vascular pathology such as vessel injury and atherosclerosis. After vessel injury, VSMCs increase their expressed of both Jagged ligands and all Notch receptors. Interestingly, Notch3 and Notch4 expression was upregulated in VSMC regions that were adjacent to ECs in the regenerating endothelium. This suggests some interaction between ECs and VSMCs, which Lindner et al. propose play a role in VSMC regulation [56]. However, Li et al. suggest Notch1 is more important in this function than Notch3, at least after carotid artery ligation [57]. Likewise, Notch1 inhibition via siRNA has been demonstrated to inhibit VSMC growth during vessel repair [58].

More specifically to atherosclerosis, it is currently understood that Notch plays a role in regulating EC function, and the dysfunction of ECs is one of the early stages in plaque formation [59]. Studies have shown that Notch signaling in ECs is affected by a high fat diet and pro-inflammatory cytokines, both of which are generally present in atherosclerosis [59,60]. Briot et al. show that a high fat diet, TNF and IL-1β treatment all reduce Notch1 expression in ECs, and this led to an increase in monocyte binding, thereby progressing disease pathogenesis [60]. Quillard et al. similarly use TNFα and IL-1β to demonstrate an upregulation of Notch2 and a downregulation of Notch4 in ECs, which promoted apoptosis [59].

In the context of mural cells, Keuylian et al. used Notch inhibitors to reduce Notch3 expression in VSMCs, which was shown to lower SMA and SM22α expression, suggesting a phenotypic switch to a more synthetic state [61]. Similarly, in *Notch3^-/-^* mice and smooth muscle specific *RBPJκ^-/-^* mice, it was shown that VSMCs had decreased F-actin and ACTA2 levels, further supporting the idea that Notch3 suppression de-differentiates VSMCs [62]. Contrastingly, Morrow et al. applied cyclic strain to VSMCs and observed a decrease in Notch3 expression as well as inhibited VSMC proliferation and an increase in VSMC apoptosis. While apparently contradictory, it is possible that cyclic strain affects other pathways that can influence the response of VSMCs [63]. Thus, the numerous different Notch receptors and ligands, as well as some apparently conflicting results highlights the need for further research in this area. Potentially complicating matters further, it has been shown that Jagged1-induced activation of Notch2 suppresses proliferation and increases contractile marker expression (SMA, SMMHC) in VSMCs. However, VSMCs from diseased patients had no change in contractile marker expression, and proliferation was only suppressed in hyperproliferative cells [64]. This highlights the need for further research in using Notch as a therapeutic target.

It should also be noted that Notch signaling affects other vital cellular events during atherosclerosis progression. Fukuda et al. used an antibody to block Dll4 in *Ldlr^-/-^* mice fed a Western diet. This led to a reduction in plaque size, along with declines in monocyte chemoattractant protein-1 (MCP-1), macrophage numbers and plaque calcification. They also state that blocking Dll4 reduces the inflammatory macrophage phenotype [65]. However, given our knowledge of plaque plasticity, this raises the question of how this would affect smooth muscle derived macrophages. Further mechanistic understanding of the Notch signaling pathway can open avenues for it to be a possible target for future therapeutic research for atherosclerosis.

### 2.5. Role of Inflammation in Atherosclerosis

We have briefly discussed the role of inflammatory signaling in atherosclerosis and the role of IL-1β in VSMCs. As such, it is generally well accepted that atherosclerosis is largely an inflammatory disease, and immune cells play a key role in plaque formation. Studies have shown that oxLDLs can induce cholesterol crystallization which potentially acts as a DAMP to activate macrophage NLRP3 inflammasomes. Activation of NLRP3 subsequently leads to the activation of caspase-1, which is responsible for cleaving and activating IL-1β (along with other cytokines) [66]. IL-1β induces a pro-inflammatory state in VSMCs as well as aids in migration and proliferation [67]. As such, research is currently being performed to ascertain the effectiveness of targeting inflammation for atherosclerosis treatments. It has been shown using antibodies that neutralizing IL-1β reduces the size of the plaque while increasing IL-10 (an anti-inflammatory cytokine) levels and shifting monocytes to a less inflammatory state [68]. Furthermore, the famous Canakinumab Anti-Inflammatory Thrombosis Outcome Study (CANTOS) investigated the effects of canakinumab on patients who had previously experienced heart attacks. Canakinumab is a drug that targets IL-1β, and it was shown to significantly reduce inflammation as well as reduce the chance of a cardiovascular event recurrence. It should be noted that the drug had no effect on LDL cholesterol levels. While it showed canakinumab was perhaps not the ideal drug (due to increased fatalities), it heavily suggested (along with other research) the viability of targeting inflammation to treat atherosclerosis [69].

### 2.6. Ischemia

Perhaps the main consequence of atherosclerosis is ischemia, which can be defined as a reduction in blood flow, and the mechanisms behind this have been briefly mentioned above. This lack of blood flow disrupts tissue homeostasis by greatly reducing nutrient delivery and waste removal from the affected tissue. Tissue-specific cells are forced to undergo more anaerobic metabolism, which causes changes in pH, ion transport and ATP production. The extent to which this damages the tissue depends on the tissue type, duration of the ischemic event, and the extent to which blood flow is reduced [70]. It should be noted that VSMCs located in different vessels around the body developmentally originate from different germ layer regions and thus may have different susceptibilities to atherosclerosis. This would mean some tissues are more prone to ischemic events than others [71]. While restoration of normal blood flow is vital for restoring homeostasis and tissue function, it also has the potential to induce further damage, which is termed ischemia-reperfusion injury. The restoration of oxygen supply is thought to increase the generation of reactive oxygen species which may cause inflammation, DNA damage and even cell death [72]. VSMCs have been shown to play a protective role in ischemia and ischemia-reperfusion injury. Ye et al. showed that co-culturing hypoxic cardiomyocytes with VSMCs resulted in suppressed cardiomyocyte apoptosis via basic fibroblast growth factor secretion and the PI3K/Akt pathway [73]. However, further research is needed to determine ways to target VSMCs for ischemia treatment.

### 2.7. Stroke

Strokes are a pathological condition that can result from atherosclerosis induced ischemia, and is caused due to the loss of blood flow to the brain, resulting in neuronal death. It should be noted that while there are different types of strokes, ischemic strokes are by far the most common [74]. Due to the varied functions of different regions of the brain, stroke victims can present a variety of different symptoms which complicates disease diagnosis. As such, stroke symptoms can be categorized into anterior cerebral artery infarction, middle cerebral artery infarction, posterior cerebral artery infarction, and cerebellar infarction, which helps in identifying the region where the ischemic event has occurred [75]. Quick disease diagnosis is essential as it is estimated that nearly 2 million neurons are lost per minute of a stroke [76]. With respect to VSMCs, strokes appear to increase the expression of endothelin-1 receptors and angiotensin II receptors in VSMCs. This leads to vasoconstriction of the vessels surrounding the affected tissue, which may worsen disease outcomes. Maddahi and Edvinsson suggest that stopping the upregulation of this receptor in VSMCs may be a good way to reduce neuronal death [77]. In addition to VSMCs, it appears that pericytes are also affected in a condition such as stroke. A brief ischemic period appears to induce pericyte dilation, which is beneficial to restoring blood flow, whereas longer ischemic events seem to contract the pericytes and further progress the disease [78,79]. It should be noted that these hypoxic conditions also induce pericyte detachment and migration, but this appears to have both positive and negative effects and is thus still a topic of debate [80,81]. Strokes are of particular interest in this review, because of their capacity to lead to neurodegeneration.

## 3. Mural Cells and Neurological Disease

Neurological disorders can be defined as diseases of the nervous system and are the second biggest killer worldwide after cardiovascular diseases. They were responsible for an estimated 9 million deaths (16.5% of total deaths) worldwide in 2016, and remain the most prominent cause of disability to date [82]. Due to the high correlation between neurological disorders and age, the incidence of these diseases is expected to increase in the coming decades as life-expectancy rises. Neurodegenerative diseases are a subset of this and are characterized by a progressive loss of neurons rather than a singular loss of neuronal event [83]. Some of the more common neurodegenerative diseases include Alzheimer’s disease (AD), Parkinson’s disease (PD), Huntington’s disease (HD), Lewy body dementia (LBD) and amyotrophic lateral sclerosis (ALS).

Blood vessels play a key role in cerebral homeostasis and are responsible for delivering nutrients, such as oxygen and glucose, while removing waste from the brain. It is estimated that the brain accounts for only 2% of body weight and consumes 20% of the body’s oxygen [84]. These vessels form a tightly regulated network made up of a variety of cells, including VSMCs, pericytes and ECs. It should therefore be no surprise that many neurological disorders are caused by the disruption of cerebral blood flow. For example, a sudden drop in cerebral blood flow can cause an ischemic stroke [85], weakening of the vessel walls can cause cerebral aneurysms [86], and plaque formation can lead to intracranial atherosclerosis [87]. However, little is known about how blood vessels affect amyloid fibrils in neurodegenerative diseases.

The key to developing therapeutic treatments for neurodegenerative diseases relies on the understanding and study of the molecular mechanisms behind them. Some of these diseases are genetic such as HD [88], while others such as AD and PD appear more sporadically, and thus their mechanisms have been harder to establish [89,90]. Despite this, genetic analysis of these diseases have suggested that protein misfolding and their subsequent aggregation may be an underlying cause [91]. This underlying mechanism is briefly described as follows. Misfolding of proteins can lead to structures that are more prone to aggregation. These proteins then interact with other misfolded proteins to form oligomers. While these early stages are unfavorable thermodynamically, oligomers rapidly associate with each other and elongate to form highly stable amyloid fibrils. They have the capacity to fragment themselves and further propagate their growth [92,93,94]. These amyloid fibrils are insoluble, have a characteristic β-sheet structure, and are correlated with the progression of many neurodegenerative diseases [95,96]. Despite this correlation, it has been more recently suggested that the oligomer intermediates are in fact the toxic species responsible for neurodegeneration rather than the amyloid fibrils themselves [92].

### 3.1. Pericyte Dysfunction and Neurodegenerative Disease

When discussing neurodegenerative diseases, a distinction between VSMCs and pericytes needs to be made. While VSMCs are located in the vessel wall of arteries and arterioles, pericytes are their equivalent in the capillaries and venules [97]. Pericytes are crucial for the development and maintenance of the blood–brain barrier (BBB), which mediates the delivery of nutrients to, and removal of waste from the central nervous system (CNS) [98,99]. Furthermore, pericyte secretions have been shown to be involved in neural stem cell differentiation and proliferation within the neural stem cell niche [100]. The distinction between these vascular cells is important because recent studies suggest that it is the relaxation and contraction of pericytes that regulates most of the changes in cerebral blood flow rather than VSMCs [101]. While there has been some debate regarding this statement [102,103], it seems the conflict stems from the nomenclature and how pericytes have been defined. Atwell suggests using a more historical definition, which includes multiple subclasses of pericytes [97]. Pericytes closer to the arterioles and venules express more smooth muscle α-actin (α-SMA) and could therefore be more involved in blood flow regulation. Pericytes in the middle of the capillaries lack α-SMA and may be more involved in BBB maintenance [104]. Currently, the molecular markers used for pericyte identification are: platelet-derived growth factor receptor-beta (PDGFR-β), chondroitin sulfate proteoglycan 4 (NG2), alanyl aminopeptidase (CD13), α-SMA (depending on pericyte location), and desmin. However, it should be noted that none of these markers are purely pericyte specific, and thus a combination of markers is usually necessary [105]. Nevertheless, the BBB and cerebral blood flow, and thus by extension pericytes, appear to be heavily involved in the pathogenesis of neurodegenerative diseases [106]. The two proposed, and probably not distinct, mechanisms of neurodegeneration are as follows. Pericyte dysfunction leads to the breakdown of the BBB, which in turn leads to the accumulation of toxic proteins in the brain. Pericyte dysfunction also leads to the reduction in cerebral blood flow, which reduces nutrient and oxygen delivery to the brain, causing secondary neurodegeneration [9].

### 3.2. Role of Mural Cells in Alzheimer’s Disease

AD is the most common neurodegenerative disease worldwide, characterized histologically by the presence of extracellular amyloid-β (Aβ) plaques and intracellular neurofibrillary tangles (NFTs), containing hyperphosphorylated tau proteins [107].

Multiple studies have shown that patients with AD or vascular dementia have more permeable BBBs than regular aging controls [108]. The BBB plays a crucial role in removing Aβ protein from the brain [109]. Sengillo et al. demonstrated that pericyte coverage of brain capillaries dropped in AD by around 30%, using both PDGFR-β and CD13 markers [110]. This correlation between pericyte loss and BBB permeability is highly suggestive that pericyte loss in AD is responsible for the rise in BBB permeability. Pericyte loss and its influence on Aβ plaque formation was investigated by Sagare et al. [9]. Overexpression of the Aβ precursor protein (APP) in pericyte deficient mice resulted in an acceleration of Aβ plaque formation. Furthermore, it was shown that pericytes express low-density lipoprotein receptor related protein 1 (LRP1) in order to clear Aβ, but too much Aβ can lead to the loss of pericytes. Thus, they have indicated that a loss in pericyte number during AD leads to the accumulation and reduced clearance of Aβ proteins, which in turn reduces the pericyte population, thus creating a cascade effect. Pericyte and/or VSMC loss may also contribute to the drop in cerebral blood flow and pulse pressure that has been shown to occur in AD (Figure 3) [111]. This drop in cerebral blood flow has been hypothesized to reduce nutrient delivery and waste removal, increase oxidative stress and Aβ aggregation, which all lead to an overall decline in neuronal cells [112,113].

While environmental factors appear to have a large contribution to AD (although exact causes have yet to be identified), there are some genes that have been associated with, and suspected to increase the susceptibility to AD. Genome wide association studies have identified multiple genes associated with AD; Apoe, ATP-binding cassette subfamily A member 7 (ABCA7), App, Presenilin 1 (Psen1), and Presenilin 2 (Psen2) [114,115,116]. Apoe has multiple isoforms, with the ε4 allele (Apoe4) having been identified in both early and late-onset AD patients. Apoe4 encodes a form of the APOE protein that is thought to bind to very low-density lipoproteins and reduce the clearance of Aβ in comparison to the other APOE isoforms [117]. ABCA7 loss of function mutations lead to increased β-secretase levels, which promotes Aβ formation [118]. App is associated mainly with familial early onset AD, and many of the associated App mutations lead to changes in Aβ expression. This can be a simple upregulation of Aβ expression, or can affect the APP cleavage sites which in turn generate more aggregation prone forms of Aβ [114,119]. Psen1 and Psen2 both encode protein components of γ-secretase, which is involved in the cleavage of the APP. Mutations in these genes alter the cleavage of APP and thus increase aggregation prone forms of Aβ [114]. It should be noted that these are just some of the most commonly seen genes that are associated with AD, and there may be rarer mutations that also contribute to AD pathogenesis, such as Notch3 [120,121].

### 3.3. Role of Mural Cells in Parkinson’s Disease

PD is another common neurodegenerative disease, which is characterized histologically by the presence of intracellular Lewy bodies, containing αSyn fibrils [122]. These Lewy bodies are thought to lead to the cell death of dopaminergic neurons in the substantia nigra, which in turn affects motor control and produces the Parkinsonian tremors and gait. However, the exact mechanism by which the fibrils achieve this is not completely understood and is being researched [123].

Compared with other neurodegenerative diseases, the role of pericytes and VSMCs in PD progression is less well known. Tunneling nanotubes (TNTs) are actin-based tunnels between cells that facilitate cell signaling and the transfer of other proteins [124,125]. These TNTs have been proposed as a mechanism by which intracellular protein aggregates can spread and progress neurodegenerative diseases [126,127,128]. Dieriks et al. were able to observe the uptake of αSyn by pericytes that were nearby neurons in the olfactory bulb, and further observed the transfer of αSyn between pericytes [127]. Thus, these results suggest that pericytes may play an important role in the αSyn spreading hypothesis. The other potential mechanism by which αSyn can spread is via secretion and subsequent uptake. However, further study is needed to determine the underlying dominant mechanism or if they work in conjunction [10]. Gray et al. looked at the effect of PD on BBB permeability post-mortem and were able to determine a loss in BBB integrity [129]. It has been suggested that monomeric αSyn may be responsible for affecting pericytes and the BBB breakdown [130].

Genetics also plays a role in PD pathogenesis. While it may affect early onset PD more, sporadic PD has still been associated with certain genes. Some of these genes include; Snca, Lrrk2, Prkn, Pink1, Dj-1, and Atp13a2 [131]. Snca is the gene that encodes the αSyn protein, and mutations in this gene that cause PD affect the N terminus, which seems to help form more stable β-sheet structures and thus promote fibril formation [131,132]. Lrrk2 encodes a protein kinase that normally helps regulate microglia and their phagocytosis of αSyn. Mutations induce higher LRRK2 activity, which reduces microglial clearance of αSyn [133,134]. Prkn encodes an E3 ubiquitin ligase and mutations generally lead to its loss of function. These mutations appear to lead to the accumulation of aminoacyl-tRNA synthetase interacting multifunctional protein type 2 and far upstream element binding protein 1, but how these induce PD progression is still to be determined [135]. Pink1 encodes a protein kinase responsible for PRKN recruitment [131]. Dj-1 is heavily involved in preventing oxidative stress as well as regulating the immune system [131,136]. Atp13a2 still requires lots of further study, but it seems to be involved in degrading αSyn aggregates [131,137]. While still preliminary, many of these PD associated genes may also be involved in the pathology of cardiovascular diseases [138,139,140].

### 3.4. Mural Cell Plasticity in Neurodegenerative Diseases

Compared with VSMC plasticity far less is known about the plasticity of pericytes, and even less of this research is specific to neurological disorders. Nevertheless, given the similarity in function and markers that VSMCs and pericytes share, it would not be surprising if pericytes too had a large degree of phenotypic variation.

He et al. performed sc-RNA seq and defined 1 pericyte cluster and 3 VSMC clusters in the 15 total brain vessel cell clusters [141]. Further study of the mural cell clusters demonstrated the brain vasculature heterogeneity, suggesting that gene expression changes throughout the vasculature. Mural cells in and closer to the arteries express more ACTA2 and SM22α, whereas cells in and closer to the venous end express more ABCC9 [142].

Pericytes are relatively undifferentiated mural cells that can be isolated from several organs. Their physiological state has been likened to that of mesenchymal stem cells (MSCs) and, indeed, it has been hypothesized that these cells may account for the perivascular origin of MSCs. This has been demonstrated by the shared expression of commonly used MSC surface markers such as CD44, CD73, CD90 and CD105, by native perivascular/pericyte cells. In addition, long-term culturing of pericytes/perivascular cells in appropriate differentiation conditions produced osteocytes, chondrocytes, and adipocytes, which could be described as a fundamental property of MSCs [143]. Thus, pericytes may progressively differentiate along distinct lineages depending on its microenvironment and its physiological state. It is worthwhile to note that a study by Guimaraes-Camboa et al. produced contradictory results. Here, the authors propose that pericytes (defined as Tbx18+ cells) do not contribute to distinct lineages in vivo in physiological or pathological models, despite such transdifferentiation in in vitro models [144]. This suggests some fundamental mechanism present in the body that we have yet to identify, which limits pericyte plasticity in vivo. Identifying this mechanism could potentially be useful if we want to transdifferentiate pericytes into helpful phenotypes to fight disease.

Further studies in vivo seem to suggest that pericytes have the potential to transdifferentiate into microglia-like cells (the equivalent of macrophages in the CNS) and other cells via a multipotent stem cell intermediate step. Using ischemic stroke mouse models, Sakuma et al. demonstrated that brain pericytes can start to express Iba1 and CD206 (microglial markers) a few days post stroke. As with the macrophage-like VSMCs, it was demonstrated that microglia-like pericytes had phagocytic abilities. Furthermore, they found that some pericytes gained stem cell-like capabilities, through the expression of nestin, c-myc, Klf4 and Sox2 (stem cell markers). These pericyte-derived stem cells had the capability of differentiating into neuronal-like, osteoblast-like and adipocyte-like cells. Interestingly, these pericyte-derived stem cells could only be isolated under ischemic conditions, thus contributing to the idea that pericyte plasticity is limited under physiological conditions [145].

Despite this, very little is known about the mechanisms by which transdifferentiation is induced in pericytes. It has been suggested that the pluripotency factors Klf4 and Oct4 may play a role, but these experiments have not been performed on brain pericytes specifically [146,147]. Interestingly, Klf4 appears to play a role in AD via inflammatory mechanisms, but it should be emphasized that the involvement of pericytes still remains unknown [148].

In the context of neurodegeneration, ischemic conditions induced by reduced blood flow could therefore lead to the phenotypic modulation of pericytes to microglia-like cells, which could contribute to disease progression. We have previously discussed the loss of pericytes in AD and PD, and an increase in microglia could contribute to the neuroinflammatory state present in both diseases.

### 3.5. PDGF Signaling in Neurodegenerative Diseases

In both AD and PD, we have discussed the loss in BBB function, which has been suggested to be a result of impaired pericyte regulation. Specifically to AD, research has suggested that Aβ accumulation can cause the loss of pericytes to accelerate neurodegeneration. However, the question related to the effect on pericytes is still not completely understood. One possible mechanism is via PDGF signaling. PDGF signaling is extensively used by neurons and glial cells during development. While there are a variety of PDGF ligands and receptors, we will focus on the PDGF-BB ligand and PDGFR-β receptor that is expressed by ECs and pericytes/VSMCs, respectively [149]. Pdgf-b^-/-^ mice had no pericytes in their vasculature, which led to microaneurysms rupturing during gestation [150]. Heterozygous Pdgfr-β mice showed a reduction in pericyte number, which also suggested that pericyte number is correlated with PDGFR-β expression [151]. Furthermore, it has been shown in AD that Aβ and hypoxia can induce the shedding of PDGFR-β from pericytes, but not VSMCs. This was associated with pericyte injury and subsequent BBB damage [152]. Thus, altered PDGF signaling could be responsible for the reduction in pericyte number that is seen in many neurodegenerative diseases and could therefore be a target for future therapeutic research.

### 3.6. Notch Signaling in Neurodegenerative Diseases

Another important mechanism could be through Notch signaling, which is a possible pathway involved in the pericyte reduction seen in these neurodegenerative diseases. It is known to be involved in a plethora of cell types and tissues, acting from development to the maturation stage. In the CNS, Notch signaling plays a crucial role in controlling the differentiation, proliferation and survival of neural stem cells, as well as maintaining neurons and regulating their remodeling [153]. As mentioned earlier, *Notch3* is a potential gene that could be involved in the pathogenesis of AD and other similar neurodegenerative conditions. *Notch3* mutations are perhaps most well-known in cerebral autosomal dominant arteriopathy with sub-cortical infarcts and leukoencephalopathy (CADASIL). CADASIL is a type of vascular dementia which causes ischemic strokes and a progressive cognitive decline [154]. Over 200 *Notch3* mutations have been found to be associated with CADASIL, but it should be noted that we currently do not know if loss of function mutations are causative as well [155]. *Notch3* is expressed by pericytes and VSMCs, and *Notch3^-/-^* mice have shown a progressive decline in VSMCs, which is similar to what is seen in CADASIL. Henshall et al. showed that this knockout leads to the disruption and leakage of the BBB, but suggest that the knockout has no effect on the pericytes [156]. In contrast, Wang et al. showed that Notch3 was necessary for pericyte proliferation in the zebrafish brain [157]. These contradictory results indicate that further research is necessary to understand the effect of Notch3 on pericytes. Despite this, Notch1 and Notch3 have been shown to regulate PDGFR-β expression, whereby *Notch3^-/-^* mice have reduced PDGFR-β levels [158,159]. Additionally, PDGFR-β is reduced in people affected by CADASIL [159]. This suggests that these signaling pathways presented may not be entirely distinct, and can influence each other in neurodegeneration pathogenesis.

It should also be noted that Notch is involved in neurodegeneration via non-pericyte pathways as well. For example, Psen1 is not only involved in cleaving APP, but also Notch1 to separate the Notch1 intracellular domain. If APP levels rise, Notch1 gets cleaved less, resulting in less Notch signaling [160]. Interestingly, it has been shown that γ-secretase (of which Psen1 is a part of) activity changes to produce more aggregate-prone forms of Aβ as we age, and that females have higher activity levels, which may explain why age and gender (female) are risk factors for AD [161].

### 3.7. Role of Neuroinflammation In Neurodegenerative Diseases

Growing evidence suggests that neuroinflammation may play a significant role in the pathology of neurodegenerative conditions. Inflammatory states have been seen in AD and PD patients, along with post-mortem tissue samples [162,163]. At the moment, the relationship between neuroinflammation and amyloid fibril accumulation is not fully understood. It is currently hypothesized that protein aggregation is sensed by microglia, which can induce an inflammatory response [164]. In support of this hypothesis, Saresella et al. showed that Aβ stimulation of monocytes in AD patients caused an activation of NLRP3, along with subsequent increases in related cytokines [165]. Similarly, Codolo et al. demonstrated that αSyn fibrils upregulated NLRP3 expression and subsequently increased IL-1β release. This study showed that the monomeric αSyn protein did not activate NLRP3 [166]. Interestingly, Quan et al. showed that NLRP3 is needed for maintenance of the pericyte population. This appears contradictory as we know NLRP3 is activated during AD and PD, which in turn have diminished pericyte numbers. Quan et al. hypothesize the involvement of some unknown mechanism which is not currently explored [167]. This highlights the gap in our current understanding with regard to neuroinflammation and neurodegenerative diseases.

## 4. Conserved Therapeutic Targets: A Clinical Perspective

Mouse models play an important role in determining the molecular mechanisms underlying these CVDs and neurodegenerative diseases, as well as provide crucial information about potential therapies. Table 1 summarizes some of the more common disease mouse models. It should be noted that PD models are still far less developed in comparison to AD models. While Table 1 shows the genetic models of PD, drugs such as 6-hydroxydopamine (6-OHDA) and 1-methyl-4-phenyl-1,2,3,6-tetrahydropyridine (MPTP) may improve the pathogenic model [168].

There is a significant synergy between the key signaling mechanisms that are utilized by cardiovascular disease and neurodegenerative disease. Understanding the possible implications of these shared signaling pathways in the context of cardiovascular and neurological disorders and understanding their synergistic relevance can lead to potential therapeutic and clinical outcomes.

As aforementioned, PDGF signaling has important roles in maturation of vascular walls and mural cell recruitment in the latter stages of angiogenesis. It has strong ties to atherosclerosis whereby an upregulation of PDGF signaling results in the promotion of plaque formation and an increase in plaque burden [50]. In contrast, in a neurovascular context, current studies suggest downregulating PDGF signaling is associated with pericyte loss and subsequent BBB dysfunction [151]. Imatinib, a first line chemotherapeutic treatment for chronic myeloid leukemia, is a selective tyrosine kinase inhibitor of PDGF signaling, among other pathways including FLT3, Lck and MAPK cascades [180]. Several preliminary observational reports suggest a beneficial role for imatinib in the management of atherosclerosis through attenuation of plaque burden and vascular dysfunction [181]. Interestingly, some preclinical models on AD suggest that imatinib also imparts a protective effect against neurodegeneration through decreases in circulating levels of Aβ and hyperphosphorylated tau [182]. It has been suggested that the efficacy of imatinib is mediated through altered processing of APP by β-secretase enzymes, leading to the reduced formation of β-CTF that is the precursor of Aβ [182]. Gianni et al. demonstrated that processing of APP is regulated by PDGF-BB through an Src-Rac dependent pathway [183]. On the other hand, imatinib treatment did not lower levels of Aβ in chronic myeloid leukemia patients [184]. As such, there is still a debate about the potential of imatinib for AD treatment. It should be noted that imatinib not only inhibits PDGF signaling but also Abl tyrosine kinase and c-kit [185]. Additionally, PDGF signaling does express many cell types including mural cells. The off-target effects may therefore be responsible for the mixed results that have been found.

Similarly, Notch signaling, has been shown to play a crucial role in tissue development and, of more relevance to our discussion, tissue homeostasis [51]. It has been described as a negative regulator of VMSC dysfunction and inhibits early atherosclerotic plaque development [186]. Results regarding its effect on pericytes have been mixed and further research is required in this niche [155,156]. Our understanding of Notch signaling in the setting of vascular dysfunction is preliminary, and there are currently no drugs, investigational or otherwise, that target this pathway [186]. However, Notch signaling knockout studies have highlighted its importance in the pathogenesis of atherosclerosis and AD, and, thus, could provide valuable therapeutic targets for clinical application [156,157].

As highlighted in the preceding sections, inflammatory signaling pathways adversely promotes the disease progression of both cardiovascular and neurodegenerative disease [68,162]. Despite substantial laboratory evidence supporting this, clinical data has produced very mixed results at best regarding the efficacies of different anti-inflammatories. Long-term use of nonsteroidal anti-inflammatory drugs (NSAIDs) and selective COX-2 inhibitors (coxibs) did not prove beneficial against atherosclerosis, with certain classes, such as ibuprofen, diclofenac, rofecoxib and lumiracoxib increasing the risk of myocardial infarction and stroke [187]. Similarly, long-term use of NSAIDs did not produce many promising results during clinical trials in patients with AD [188]. The CANTOS trial highlighted the need for more target anti-inflammatories, particularly through the utilization of monoclonal antibodies against specific mediators of inflammation. Of particular interest are IL-1β and TNF cytokines that are, perhaps, the most potent pro-inflammatory mediators. The use of canakinumab, an anti-IL-1β monoclonal antibody, in the context of atherosclerosis, has been promising, however, not enough data is available on its use in AD [69]. Long-term TNF blockade reduced cardiovascular incidents in several observational studies of patients with arthritis [189]. In addition, TNF blockade via the use of etanercept has shown improved cognition in patients with AD [190]. It should be noted that these results are based on small sample sized clinical studies that lacked a placebo group. Thus, a larger placebo-controlled study would be needed to verify the efficacy of TNF blockade. In summary, further investigation is needed to elucidate the potential efficacy of IL-1β and TNF inhibition as a therapeutic strategy against neuroinflammation and, by extension, against neurodegeneration.

## 5. Conclusions

The role of the vasculature in the onset of CVD is a well-established concept. Conversely, its role in the pathogenesis of neurodegeneration has been a more recent exploration [108]. Due to a wealth of laboratory data highlighting the importance of cerebral blood flow and hypoxia in the pathogenesis of neurodegeneration, the role of vasculature, particularly of vascular smooth muscle cells and pericytes, has become an increasingly important focus of research for novel therapeutic targets.

In this review, we have highlighted three key signaling pathways that act on mural cells, shared by both cardiovascular and neurodegenerative diseases: PDGF signaling, Notch signaling and inflammatory signaling. Preliminary clinical studies on drugs that act on these pathways in the domain of cardiovascular disease management appear to impart protective qualities against dementia in addition to its cardioprotective actions.

A few key questions do need to be addressed prior to utilizing mural cells as a therapeutic target. Due to the association between CVD and neurodegeneration with the former preceding the latter, a precise understanding of these disease processes on a temporal framework needs to be established. In essence, do the aforementioned mural cell pathologies relevant to neurodegeneration occur within the same time frame as those relevant to atherosclerosis? Literature on the initiation and progression of atherosclerosis suggests that these disease processes can begin and progress decades prior to overt clinical manifestations. For example, diffuse intimal thickenings are widely considered the precursors to atherosclerotic plaques and have been found from birth [191]. However, the literature on the temporal framework of mural cell pathologies associated with neurodegeneration is relatively sparse. Furthermore, do the disease processes change over time and to what extent does this affect the clinical efficacy of therapeutics? A prime example of the dynamic nature of these disease processes is highlighted in mouse studies conducted by Gomez et al. [192]. Their work showed that IL-1β antibody administration in late-stage atherosclerosis produced no changes in lesion size and in fact decreased beneficial outward remodeling. They suggest an atheroprotective role for IL-1β in late-stage atherosclerosis. This is in contrast to our conventional understanding that IL-1β, a harbinger of inflammation, is generally detrimental to atherosclerosis [192]. Thus, for example, despite its cardioprotective effects in early atherosclerosis, the continued administration of canakinumab in an effort to mitigate the onset of neurodegeneration could in fact worsen an individual’s atherosclerosis in their later stages of life. It is thus imperative that a clear understanding of the evolution of mural dysfunction be established.

Secondly, is there a role for mural cell plasticity in therapeutic approaches towards neurodegeneration and CVD? As highlighted in previous sections, transdifferentiation of VSMCs into osteochondrogenic-like or macrophage-like cells contribute to adverse increases in disease burden. In neurodegenerative states, transdifferentiation of pericytes to microglia-like cells may similarly contribute to an increased disruption of the BBB due to decreased pericyte coverage, and increased neuroinflammation [145]. In addition, PDGF signaling has been implicated in both plastic processes, making this signaling cascade and relevant therapeutics interesting candidates for future study [47].

Thirdly, to what extent does heterogeneity play a role in defining the origin of mural cell pathologies? It has been identified in atherosclerosis that the fibrous smooth muscle cap of an atherosclerotic lesion is derived from clonal expansion of a few VSMCs in the medial intima of the vascular wall [7]. It has been hypothesized that these cells represent a subgroup primed to respond to injury and mechanism of lateral inhibition have also been described [193]. Despite heterogeneity being established in pericyte populations, no ‘at-risk’ sub-clonal populations of pericytes have been isolated [104]. Indeed, much of this is owed to the difficulties in defining markers specific to pericytes and precise identification/isolation of sub-clonal populations [105].

It is clear that our understanding of mural cell pathology in the context of neurodegeneration is still quite limited. Clinical guidelines in the future, informed by laboratory work in this niche, could address both cardiovascular disease and subsequent onset of dementia as a single entity rather than two mutually exclusive systems. Clinicians would be able to prescribe particularly vulnerable patients (identified through genetic analysis or strong family history or clinical risk stratification) with particular classes of CVD drugs that also impart protective effects against the subsequent onset of neurodegenerative disease.

## Figures and Tables

**Figure 1 cells-10-00593-f001:**
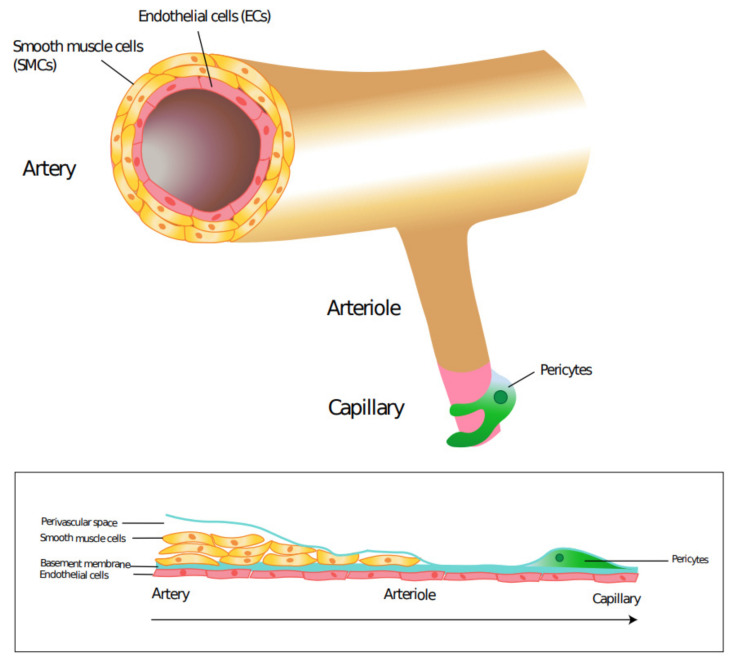
Location of pericytes and vascular smooth muscle cells in blood vessels. Smooth muscle cells continuously surround endothelial cells in arteries, arterioles, veins, and venules. Pericytes form a discontinuous layer surrounding endothelial cells in capillaries.

**Figure 2 cells-10-00593-f002:**
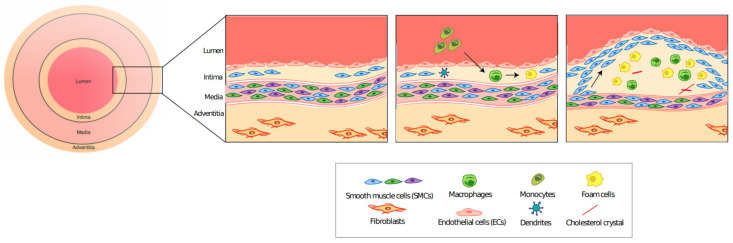
Plaque formation in atherosclerosis. Endothelial cell dysfunction leads to the recruitment of circulating monocytes, which become macrophages. Macrophages phagocytose oxLDLs to form foam cells. This inflammatory process stimulates the migration of SMC from the medial wall to form the fibrous cap.

**Figure 3 cells-10-00593-f003:**
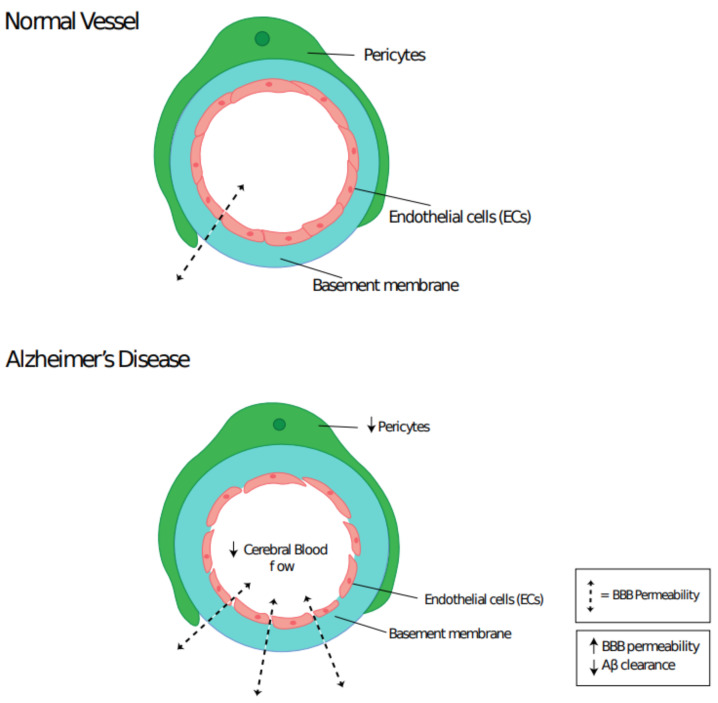
Cerebral capillary integrity is reduced in Alzheimer’s Disease pathogenesis. Pericyte loss leads to a drop in cerebral blood flow and increased blood–brain barrier permeability, which results in Aβ accumulation.

**Table 1 cells-10-00593-t001:** Some commonly used mouse models for cardiovascular diseases and neurodegenerative diseases. Diseases included are: Atherosclerosis, Ischemia, Stroke, Alzheimer’s Disease and Parkinson’s Disease.

Disease	Model	Phenotype	Reference
Atherosclerosis	*Apoe^-/-^*	High cholesterol levels	[169]
		Increased sensitivity to fat and cholesterol-based dietsExtensive atherosclerosis by 3 months	
	*Ldlr^-/-^*	Plaque development only in high fat diets	[169,170]
		Better mimics human pathogenesis (lipoprotein profile)	
		Atherosclerosis induction by 6 months	
	*Apoe^-/-^* + *Ldlr^-/-^*	More severe atherosclerosis than *Apoe^-/-^* and *Ldlr^-/-^* individual knockouts	[169]
		Lipoprotein profile similar to *Apoe^-/-^*	
Ischemia	Suture occlusion of artery	Depends on which affected tissue is being modelled	[171,172]
Stroke	Suture to occlude the middle cerebral artery	Infarction (size dependent on occlusion time, suture size, suture material, etc.)	[172]
		Striatum blood flow normalizes after 2 h	
		Cortical blood flow remains low	
	Endothelin-1 injection directly to middle cerebral artery	Infarction (size dependent on dose)	[172]
		Reduced cerebral blood flow–reperfusion takes hours	
	Microsphere insertion into the middle cerebral artery	Infarction (size dependent on microsphere size, material)	[172]
Alzheimer’s	*App-Indiana* mutation with PDGF promoter	18× APP RNA	[173,174]
		10× APP protein	
		Amyloid deposition	
		apparent at 9 months	
		Cerebral amyloid angiopathy	
	*App-Swedish* mutation with hamster prion protein promoter	5× APP protein	[9,173,175]
		Amyloid deposition apparent at 11–13 months	
		Cerebral amyloid angiopathy	
	*App-Swedish* + *Indiana* mutation with hamster prion protein promoter	Amyloid deposition apparent at 3 months	[173,176]
		Cerebral amyloid angiopathy	
Parkinson’s	*Snca-a53t* mutation with mouse prion protein promoter	Initial motor control degradation and α-Syn inclusions at 8 months	[177]
	*Snca-a30p* mutation with hamster prion protein promoter	Initial motor control degradation at 13 months	[178]
	*Snca-e46k* mutation with mouse prion protein promoter	Initial motor degradation at 16 months	[179]
		Slower disease progression than other mutations	

## Data Availability

Data sharing not applicable.

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
