# Peer review of "Mural Cells: Potential Therapeutic Targets to Bridge Cardiovascular Disease and Neurodegeneration"

_cells, 2021, doi:10.3390/cells10030593_

Round 1

Reviewer 1 Report

The review addresses the specific role of mural cells (vascular smooth muscle cells and pericytes) within the vessel wall and their contribution to the pathology of atherosclerotic and neurodegenerative disease, respectively, and further the signalling pathways (PDGF, Notch and inflammatory signaling) that are conserved across both diseases.

The role of mural cells within the vasculature in neurodegenerative disorders and vascular disease has recently been reviewed in Nat Neurosci. 2018; 21(10): 1318–1331; doi: 10.1038/s41593-018-0234-x. and Nat Rev Cardiol, 2019;16(12):727-744; doi: 10.1038/s41569-019-0227-9 and Cardiovac Res, 2018;114(4):477–480; https://doi.org/10.1093/cvr/cvy031, respectively. This review brings together these two fields of research into focus to establish whether specific similarities exist for these co-morbidities.

Major points:

1.     Single cell RNA sequencing (scRNAseq) studies of mural cells in normal and diseased vessels suggests a heterogeneity within the vessel wall - these studies should be cited and evaluated (Kalluri et al; 2019; DOI: 10.1161/CIRCULATIONAHA.118.038362,  Vanlandewijck et al 2018; https://doi.org/10.1038/sdata.2018.160 and  Alencar et al., 2020; https://doi.org/10.1161/ CIRCRESAHA.120.316770). These recent studies provide evidence that mural derived cells within advanced mouse and human atherosclerotic lesions exhibit far greater phenotypic plasticity than generally believed, with Klf4 regulating transition to multiple phenotypes including Lgals3+ osteogenic cells likely to be detrimental for late-stage atherosclerosis plaque pathogenesis. This new information should be included.

  • The overall role of mural cells in atherosclerotic disease is controversial with many studies now suggesting that it’s a a rare population of Sca1/Myh11-Cre marked medial cells contribute to intimal thickening - these studies should be cited (Dobnikar et al., https://doi.org/10.1038/s41467-018-06891-x ; Tang et al., 2020; doi: 10.1016/j.stem.2019.11.010).  Lineage tracing studies had previously suggested that a specific clonal subpopulation of mural cells were responsible (Chappel et al., Circ Res. 2016; DOI: 10.1161/CIRCRESAHA.116.309799).
  • In addition, the potential expansion of resident vascular stem cells in contributing the mural cell infiltration, intimal thickening and lesion formation for several  vascular pathologies should also be addressed as many of the lesional cells may be derived from this source (Kramann et al., 2016; doi: 10.1016/j.stem.2016.08.001, Tang et al., 2020 doi: 10.1016/j.stem.2019.11.010, 2012 doi: Tang et al., 2012; 10.1038/ncomms1867.; Yuan et al., 2017; doi: 10.1371/journal.pone.0168914 ). The potential role of Notch signaling in dictating mural cell phenotype has been widely published where Notch ligands promote myogenic differentiation of undifferentiated stem cells.
  • The section on the contributory role of Notch signalling in both lesion formation and neurodegenerative disease is somewhat superficial and should be expanded to include more mechanistic studies that addressed the among other things the role of individual receptors Notch 1-4 and target genes in mural cells (e.g., Am J Pathol. 2001; 159: 875–883; doi: 10.1016/S0002-9440(10)61763-4, Circulation, 2009 119: 2686–2692; doi: 10.1161/CIRCULATIONAHA.108.790485, PLoS One. 2014; 9(1): e84122; doi: 10.1371/journal.pone.0084122) following vascular injury, and within the neurovascular unit/network (https://doi.org/10.1016 /j.semcdb.2020.12.011) and JCI Insight 2020;5(21):e125940
  • The authors should expand on the major milestones in basic and clinical research that need to be reached before therapies that target the role of      mural cells in both pathologies need to be established.

Minor Points;

  • The authors should cite the primary literature unless referring to a specific review article.

Author Response

Dear Reviewer,

Thank you for taking the time to read and provide feedback on our review.  It was extremely useful, and we hope these revisions have addressed your concerns.

Single cell RNA sequencing (scRNAseq) studies of mural cells in normal and diseased vessels suggests a heterogeneity within the vessel wall - these studies should be cited and evaluated (Kalluri et al; 2019; DOI: 10.1161/CIRCULATIONAHA.118.038362,  Vanlandewijck et al 2018; https://doi.org/10.1038/sdata.2018.160 and  Alencar et al., 2020; https://doi.org/10.1161/ CIRCRESAHA.120.316770). These recent studies provide evidence that mural derived cells within advanced mouse and human atherosclerotic lesions exhibit far greater phenotypic plasticity than generally believed, with Klf4 regulating transition to multiple phenotypes including Lgals3+ osteogenic cells likely to be detrimental for late-stage atherosclerosis plaque pathogenesis. This new information should be included.

We have added sections on VSMC and pericyte plasticity.  In the VSMC section (pages 4-6) we discuss transdifferentiation of VSMCs to SM-derived macrophage-like and SM-derived osteochondrogenic-like states.  We also mention the scRNA seq data and the overall heterogeneity within the atherosclerotic plaque.  As suggested, we also briefly look into Klf4 and Oct4 as potential mechanisms behind transdifferentiation.  In the pericyte section (page 12) we discuss the heterogeneity of mural cells within the brain.  Compared to atherosclerosis, pericyte plasticity is far less established in neurodegeneration.  We discuss the role of pericytes as potential origins of mesenchymal stem cells, as well as their capability to transdifferentiate into microglia-like cells.  Similarly, we briefly mention Klf4 and Oct4 in the context of neurological disorders, but not enough is known about how they regulate brain pericytes.

The overall role of mural cells in atherosclerotic disease is controversial with many studies now suggesting that it’s a a rare population of Sca1/Myh11-Cre marked medial cells contribute to intimal thickening - these studies should be cited (Dobnikar et al., https://doi.org/10.1038/s41467-018-06891-x ; Tang et al., 2020; doi: 10.1016/j.stem.2019.11.010).  Lineage tracing studies had previously suggested that a specific clonal subpopulation of mural cells were responsible (Chappel et al., Circ Res. 2016; DOI: 10.1161/CIRCRESAHA.116.309799).

In addition, the potential expansion of resident vascular stem cells in contributing the mural cell infiltration, intimal thickening and lesion formation for several  vascular pathologies should also be addressed as many of the lesional cells may be derived from this source (Kramann et al., 2016; doi: 10.1016/j.stem.2016.08.001, Tang et al., 2020 doi: 10.1016/j.stem.2019.11.010, 2012 doi: Tang et al., 2012; 10.1038/ncomms1867.; Yuan et al., 2017; doi: 10.1371/journal.pone.0168914 ). The potential role of Notch signaling in dictating mural cell phenotype has been widely published where Notch ligands promote myogenic differentiation of undifferentiated stem cells.

We have added a section on VSMC clonality (page 4) in the formation of the atherosclerotic plaque.  Here we discuss that one or two VSMCs clonally expand to form all SM-derived plaque cells.  However, it has been found that there are Sca1 marked VSMCs that are more proliferative and plastic (and are more involved in vascular repair), suggesting that these may be the cells that clonally expand.  It has also been suggested that because not all VSMC derived plaque cells express Sca1, and yet come from a common medial VSMC, that Sca1 is an intermediate transition state marker. 

The section on the contributory role of Notch signalling in both lesion formation and neurodegenerative disease is somewhat superficial and should be expanded to include more mechanistic studies that addressed the among other things the role of individual receptors Notch 1-4 and target genes in mural cells (e.g., Am J Pathol. 2001; 159: 875–883; doi: 10.1016/S0002-9440(10)61763-4, Circulation, 2009 119: 2686–2692; doi: 10.1161/CIRCULATIONAHA.108.790485, PLoS One. 2014; 9(1): e84122; doi: 10.1371/journal.pone.0084122) following vascular injury, and within the neurovascular unit/network (https://doi.org/10.1016 /j.semcdb.2020.12.011) and JCI Insight 2020;5(21):e125940

Our section on Notch signaling in atherosclerosis has been expanded (page 6).  We have provided a brief overview of the Notch signaling pathway.  We also discuss the role of Notch ligands and receptors in vascular development and injury.  For atherosclerosis, we provide the most updated literature on the role of Notch in VSMCs as well as briefly in endothelial cells and macrophages.  In the neurodegeneration section (page 13), we have looked at CADASIL and the effect of Notch on pericytes.  We also added a brief description on non-pericyte pathways – the competition between Notch1 and APP.

The authors should expand on the major milestones in basic and clinical research that need to be reached before therapies that target the role of      mural cells in both pathologies need to be established.

In the conclusion we now discuss some key requirements that need to be further understood before we can develop useful therapies (page 16-17). The timeframe of both diseases, and whether the provided treatments will have to change over time? The potential for targeting plasticity in both CVD and neurodegeneration? The potential of using mural cell heterogeneity to target specific ‘at risk’ populations?

Minor Points;

The authors should cite the primary literature unless referring to a specific review article.

We have cited some more primary literature (such as: https://doi.org/10.1006/dbio.1995.1291 , https://doi.org/10.1159/000471866 , 10.2174/156720212799297100).

Thank you for taking the time to read and provide positive feedback on our review.  It was extremely useful, and we hope these revisions have addressed your concerns and improved qualitity of the review. 

Reviewer 2 Report

In this review, the authors offer a comprehansive discussion on mural cell dysfunctions in cardiovascular diseases and neurodegeneration. Although these are two totally different diseases, the authors review the common signaling events involved in both diseases. While this connection is interesting, in my opinion, this review should focus on the roles of mural cells in neurodegneration, and integrate the relevant findings on cardiovasculr diseases into the discussion.

  1. Much of the review on cardiovascular diseases are repeating what have been widely published, which is unecessary. It should be simplified or integrated into the discussion on mural cells in PD/AD and make critical comments.
  2. Mural cells in the brain are around microvessels, and mural cells in large artery involving in atherosclerosis are quite different.What are the respective markers (transcriptional factors, surface markers, contractile markers, etc.)?
  3. Although similar signaling events in mural cells are involved in CVD and AD/PD, do these signaling events lead to the same phenotype/functional changes that cause the diaseases in the artery and brain? This direct comparion and discussion should be improved.

Author Response

Dear Reviewer,

Thank you for taking the time to read and provide feedback on our review.  It was extremely useful, and we hope these revisions have addressed your concerns.

  1. Much of the review on cardiovascular diseases are repeating what have been widely published, which is unecessary. It should be simplified or integrated into the discussion on mural cells in PD/AD and make critical comments.

While there are reviews that discuss CVD and neurodegeneration separately, there is not much literature that combines the two, especially in the context of mural cells.  We have tried to discuss more recent information (such as https://doi.org/10.1016/j.stem.2019.11.010 , https://doi.org/10.1161/CIRCULATIONAHA.120.046672 , https://doi.org/10.1172/jci.insight.125940 , https://doi.org/10.3389/fncel.2020.00276 )  that, to our knowledge, hasn’t been discussed much, especially in the same context of this review.  Furthermore, our goal was to write about mural cells as a connection between both CVD and PD/AD rather than specifically on neurodegeneration. 

  1. Mural cells in the brain are around microvessels, and mural cells in large artery involving in atherosclerosis are quite different.What are the respective markers (transcriptional factors, surface markers, contractile markers, etc.)?

We have added some more about markers, especially in the new context of plasticity (page 5-6).  Sca1 is discussed as a marker of vascular stem cell derived VSMCs.  Contractile VSMCs express SMMHC, α-SMA (among others), where SMMHC is downregulated and some ECM genes are upregulated in more synthetic states.  CD68 and Mac2 are used as macrophage markers, and osteopontin, cbfa1, osteocalcin and alkaline phosphatase are used as osteochondrogenic markers. We also discuss the difficulty identifying pericytes due no pericyte-specific markers (page 9).  Currently, markers are used in combination, such as PDGFR-β, NG2, CD13, α-SMA and desmin. The heterogeneity of mural cell expression in the brain is discussed (page 12), along with the markers used to identify pericyte to mesenchymal stem cell (CD44, CD73, CD90, CD105) and microglia-like (Iba1, CD206) transitions. 

  1. Although similar signaling events in mural cells are involved in CVD and AD/PD, do these signaling events lead to the same phenotype/functional changes that cause the diaseases in the artery and brain? This direct comparion and discussion should be improved.

For the signalling sections, we looked at how the different signaling mechanisms affected VSMCs in CVD, and how they affected pericytes in AD/PD.  As they are different cells in different locations, we cannot really make a direct comparison between them. 

Thank you for taking the time to read and provide positive feedback on our review.  It was extremely useful, and we hope these revisions have addressed your concerns and improved qualitity of the review. 

Reviewer 3 Report

This review on the role of wall cells in cardiovascular and neurodegenerative diseases is rather well documented. The first two parts provide an overview of the literature in the field.
The major criticism I can offer concerns the iconography.

Indeed, if the legend in figure 1 is correct, the representation made of it suggests that capillaries are composed of endothelial cells, smooth muscle cells AND pericyte. In my opinion, this figure should be modified as being made up of a monolayer of endothelial cells surrounded by a few pericytes. 

In figure 2, an unleaded T-cell receptor is shown. This should be added in the legend and in the text if it is important for the process.

inally, figure 3 must also be modified. Indeed, the  BBB is located in the cerebral capillaries, which are made up of a single layer of endothelial cells. In this figure, 2 layers are shown. It is therefore necessary to remove one of them. 

Moreover, in the pathological case of AD, the permeability of this barrier can be modified. Some receptor and/or transporter expressions are modulated and lead to a greater accumulation of AB peptide. This point is  discussed in the text. On the other hand, at the level of representation, one has the impression that the barrier is completely broken, whereas there is only a modulation of the permeability. Please modify this figure.

The third part evaluates therapeutic strategies that could be common to treat these pathologies.
The table presenting the different models is of no interest in this part. It would be more interesting to formalise a table with the different possible strategies.

In the text, other proteins from GWAS such as ABCA7 for example should be added.

Author Response

Dear Reviewer,

Thank you for taking the time to read and provide feedback on our review.  It was extremely useful, and we hope these revisions have addressed your concerns.

Indeed, if the legend in figure 1 is correct, the representation made of it suggests that capillaries are composed of endothelial cells, smooth muscle cells AND pericyte. In my opinion, this figure should be modified as being made up of a monolayer of endothelial cells surrounded by a few pericytes.

We have adjusted this figure (page 2) so that the artery, arteriole and capillary are more distinct, thereby hopefully making it clearer that SMCs are in arteries and arterioles whereas pericytes are in capillaries.

In figure 2, an unleaded T-cell receptor is shown. This should be added in the legend and in the text if it is important for the process.

We have adjusted this figure (page 4) to remove the T cell receptor.

inally, figure 3 must also be modified. Indeed, the  BBB is located in the cerebral capillaries, which are made up of a single layer of endothelial cells. In this figure, 2 layers are shown. It is therefore necessary to remove one of them.

We have adjusted this figure (page 10) so that only 1 endothelial layer is present.

Moreover, in the pathological case of AD, the permeability of this barrier can be modified. Some receptor and/or transporter expressions are modulated and lead to a greater accumulation of AB peptide. This point is  discussed in the text. On the other hand, at the level of representation, one has the impression that the barrier is completely broken, whereas there is only a modulation of the permeability. Please modify this figure.

In the new figure (page 10), we have represented the BBB permeability with gaps in the endothelial cells rather than a completely broken barrier.

The third part evaluates therapeutic strategies that could be common to treat these pathologies. The table presenting the different models is of no interest in this part. It would be more interesting to formalise a table with the different possible strategies.

While we thank you for this comment, we believe that the different mice models are useful, especially for scientists who work in one field and may want to investigate the effects in the other field.

In the text, other proteins from GWAS such as ABCA7 for example should be added.

In the AD genetics section (page 10-11), we have added a brief description on ABCA7.  However, while interesting, this section was not designed to be a huge part of this review as it doesn’t relate as much to mural cells. 

Thank you for taking the time to read and provide positive feedback on our review.  It was extremely useful, and we hope these revisions have addressed your concerns and improved qualitity of the review. 

Round 2

Reviewer 1 Report

The authors have addressed my comments and have edited the review accordingly.

One final suggestion the authors should consider is to explicitly state that SMC 'murual' cell plasticity alone may not account for all the regenerative processes at play within vascular lesions.  Therefore they should include a sentence or two that outlines the numerous studies that demonstrate a role for resident vascular stem cells (not derived from SMCs that are present in the adventitia and medial layers), endothelial cells that undergo endothelial to mesenchymal transition (EndMT) and circulating bone-morrow derived mesenchymal stem cells.

  1. Kramann R, Goettsch C, Wongboonsin J, Iwata H, Schneider RK, Kuppe C, Kaesler N, Chang-Panesso M, Machado FG, Gratwohl S, Madhurima K, Hutcheson JD, Jain S, Aikawa E, Humphreys BD. Adventitial MSC-like Cells Are Progenitors of Vascular Smooth Muscle Cells and Drive Vascular Calcification in Chronic Kidney Disease. Cell Stem Cell 2016;19:628–642.
  2. Baker AH, Péault B. A Gli(1)ttering Role for Perivascular Stem Cells in Blood Vessel Remodeling. Cell Stem Cell. 2016.
  3. Tang Z, Wang A, Yuan F, Yan Z, Liu B, Chu JS, Helms JA, Li S. Differentiation of multipotent vascular stem cells contributes to vascular diseases. Nat Commun 2012;3:875.
  4. Yuan F, Wang D, Xu K, Wang J, Zhang Z, Yang L, Yang GY, Li S. Contribution of vascular cells to neointimal formation. PLoS One 2017;
  5. Tang J, Wang H, Huang X, Li F, Zhu H, Li Y, He L, Zhang H, Pu W, Liu K, Zhao H, Bentzon JF, Yu Y, Ji Y, Nie Y, Tian X, Zhang L, Gao D, Zhou B. Arterial Sca1+ Vascular Stem Cells Generate De Novo Smooth Muscle for Artery Repair and Regeneration. Cell Stem Cell 2020;26:81-96.e4.
  6. Cooley BC, Nevado J, Mellad J, Yang D, Hilaire C St., Negro A, Fang F, Chen G, San H, Walts AD, Schwartzbeck RL, Taylor B, Lanzer JD, Wragg A, Elagha A, Beltran LE, Berry C, Feil R, Virmani R, Ladich E, Kovacic JC, Boehm M. TGF-β signaling mediates endothelial-to-mesenchymal transition (EndMT) during vein graft remodeling. Sci Transl Med 2014;6:227ra34-227ra34.
  7. Moeen A, Yin T, L. DL. Mesenchymal Stem Cells and the Artery Wall. Circ Res 2004;95:671–676.

Author Response

Dear Reviwers,

We did required amendments as per suggestions.

Thanks,

Ashish